# Improving Effects of Hop-Derived Bitter Acids in Beer on Cognitive Functions: A New Strategy for Vagus Nerve Stimulation

**DOI:** 10.3390/biom10010131

**Published:** 2020-01-13

**Authors:** Tatsuhiro Ayabe, Takafumi Fukuda, Yasuhisa Ano

**Affiliations:** Research Laboratories for Health Science & Food Technologies, Kirin Company Ltd., 1-13-5 Fukuura Kanazawa-ku, Yokohama-shi, Kanagawa 236-0004, Japan; Takafumi_Fukuda@kirin.co.jp (T.F.); Yasuhisa_Ano@kirin.co.jp (Y.A.)

**Keywords:** β-carbonyl, dopamine, iso-α-acids, matured hop bitter acids, norepinephrine, vagus nerve

## Abstract

Dementia and cognitive decline are global public health problems. Moderate consumption of alcoholic beverages reduces the risk of dementia and cognitive decline. For instance, resveratrol, a polyphenolic compound found in red wine, has been well studied and reported to prevent dementia and cognitive decline. However, the effects of specific beer constituents on cognitive function have not been investigated in as much detail. In the present review, we discuss the latest reports on the effects and underlying mechanisms of hop-derived bitter acids found in beer. Iso-α-acids (IAAs), the main bitter components of beer, enhance hippocampus-dependent memory and prefrontal cortex-associated cognitive function via dopamine neurotransmission activation. Matured hop bitter acids (MHBAs), oxidized components with β-carbonyl moieties derived from aged hops, also enhance memory functions via norepinephrine neurotransmission-mediated mechanisms. Furthermore, the effects of both IAAs and MHBAs are attenuated by vagotomy, suggesting that these bitter acids enhance cognitive function via vagus nerve stimulation. Moreover, supplementation with IAAs attenuates neuroinflammation and cognitive impairments in various rodent models of neurodegeneration including Alzheimer’s disease. Daily supplementation with hop-derived bitter acids (e.g., 35 mg/day of MHBAs) may be a safe and effective strategy to stimulate the vagus nerve and thus enhance cognitive function.

## 1. Introduction

Due to the rapid growth of the world’s elderly population, dementia and cognitive decline have become critical worldwide public health problems. There are many different types of dementia, such as Alzheimer’s disease (AD), the most common type of dementia, vascular dementia, or dementia with Lewy bodies. Recent findings suggested that obesity or lifestyle-related diseases including type II diabetes are the key risk factors for dementia [1]. Given that fundamental therapeutic strategies for dementia have not been well established, dementia prevention via changes to daily life, such as to dietary habits, or obesity and type II diabetes prevention, has received increasing attention.

Several epidemiological studies have suggested that the moderate consumption of alcoholic beverages such as wine and beer may lower the risk of dementia. Neafsey and Collins [2] reviewed 143 published papers and concluded that light to moderate alcoholic beverage consumption (≤2 drinks/day for men, ≤1 drink/day for women) may reduce the risk of dementia and cognitive impairments. In Xu et al.’s [3] dose-response meta-analysis, a nonlinear association between alcohol consumption and the risk of dementia, with moderate alcohol consumption (≤12.5 g/day) reducing risk (lowest risk with roughly 6 g/day) and excessive drinking (≥38 g/day) elevating risk, was further revealed. Intake of alcohol itself has preventive effect by coping stress or lowering blood pressure. Specific components of alcoholic beverages may also decrease some of the risk for dementia. In particular, resveratrol, a polyphenolic compound found in red wine, has some potential dementia prevention efficacy [4,5,6]. Despite this existing focus on the impacts of resveratrol and alcohol more broadly on dementia, the neuroprotective and cognitive impacts of particular components of beer have not been investigated until recently.

Hops, the female inflorescences of hop plants (*Humulus lupulus* L.), are one of the primary ingredients in beer. They have been widely used for beer production since the 9th century to achieve its characteristic bitterness and aromas, stabilize beer foam, and provide some resistance against bacterial proliferation. Since the Middle Ages in Europe, hops have also been used as a folk medicine. The antimicrobial properties and relaxation benefits of hops are found in herbal medicine books. Initial scientific reports of the health benefits of hops, such as their sleep-inducing, anti-inflammatory, and gastrointestinal effects, were also published during the 19th and early 20th century [7].

More current scientific research has revealed that some specific components derived from hops have physiological functions. For instance, 8-prenylnaringenin, a potent phytoestrogen found in hops, was identified by Milligan et al. [8]. 8-Prenylnaringenin is structurally a prenylated flavonoid and estrogen modulator that thus exhibits some beneficial effects on menopausal and post-menopausal symptoms, bone-resorption, and tumor growth [9]. Furthermore, the physiological functions of xanthohumol, another prenylated flavonoid, have been previously reviewed by Liu et al. [10]. The anti-inflammatory, antioxidant, hypoglycemic, and anti-cancer effects of xanthohumol may be effective in the context of various diseases. Liu et al. demonstrated that the anti-inflammatory and antioxidant effects of xanthohumol might contribute to the prevention of dementia and cognitive decline. However, both 8-prenylnaringenin and xanthohumol are minor components of beer. While it is now possible to isolate and concentrate the xanthohumol as its pharmacological effective dose, and to produce functional food including beer or beverages, it may be difficult to ingest an effective dose with only moderate consumption of regular beer. Considering the harmful effects of excessive alcohol consumption, bioactive molecules contained in beer are preferable to be adequately concentrated in beer.

Recently, the physiological functions and the mechanisms of hop-derived bitter acids, which are adequately contained in regular beer, have been studied. In the present review, studies investigating the effects of hop-derived bitter acids, especially on cognitive function, will be demonstrated. In addition, unique mechanisms of cognitive improvement with hop-derived bitter acid consumption, focusing on the vagus nerve stimulation, will also be discussed. Most of these studies are performed using only bitter acid compounds, not using beer, but would be valuable for explaining the dementia-prevention effects of alcoholic beverages.

## 2. Iso-α-Acids (IAAs)

### 2.1. Characterization of IAAs

IAAs are the main bitter components of beer, which are converted from α-acids in hops during the brewing process. They are potent agonists of bitter taste receptors TAS2Rs (T2Rs) [11]. IAAs are reported to be responsible for the typical bitterness of beer [12], beer foam stability [13], and antibacterial properties [14,15]. Representative IAA compounds include isocohumulone, isohumulone, and isoadhumulone, depending on the particular acyl side chain structure present, and each compound has *cis* and *trans* isomers (Figure 1). Taniguchi et al. [16] further determined the concentration of IAAs in commercially available beers and found 16–27 mg/L of IAAs in lager-type Japanese beer and 41–64 mg/L of IAAs in India Pale Ale, which is brewed with a large amount of hops.

### 2.2. IAAs Prevent Type II Diabetes, Lipid Metabolism, and Obesity-Induced Cognitive Decline

Epidemiological studies have shown that type II diabetes, especially insulin resistance which is the representative pathology of type II diabetes, or obesity increase the risk for dementia [17,18,19]. Obesity or excess body weight negatively correlates with cognitive function and with the volume of several brain regions including hippocampus [20,21]. In animal studies, high fat diet feeding induces neuroinflammation, which leads to the cognitive decline [22]. These findings suggest that preventing type II diabetes and obesity would be effective approaches in dementia prevention.

Yajima et al. [23] investigated whether IAAs activate peroxisome proliferator-activated receptors (PPARs), which regulate fatty acid and carbohydrate metabolism [24]. PPAR agonism by IAA was evaluated via transient co-transfection assays, which detect PPAR activation with luciferase activity [25,26]. IAAs, such as isohumulone, isocohumulone, and isoadhumulone, activated PPARγ in a manner nearly equivalent to that of pioglitazone, a selective PPARγ agonist. Similarly, isohumulone and isocohumulone activated PPARα. These results demonstrate the unique properties of IAAs, which activate both PPARα and PPARγ.

The anti-diabetes and anti-obesity effects of IAAs were further examined in animal studies. For instance, treatment of diabetic KK-A^y^ mice with IAAs resulted in reduced plasma glucose, triglycerides, and free fatty acids. IAAs also improved insulin resistance and glucose tolerance in high fat diet (HFD)-fed C57BL/6 mice. Yajima et al. [27] demonstrated that dietary supplementation with IAAs improved diet-induced obesity and insulin resistance in HFD-fed mice, effects mediated by lipid metabolism and the inhibition of intestinal lipid absorption. Miura et al. [28] focused on cholesterol metabolism and revealed that dietary supplementation with IAAs elevated plasma high-density lipoprotein (HDL)-cholesterol levels and reduced liver cholesterol in HFD-fed mice.

The effect of IAAs on obesity or type II diabetes has further been confirmed in clinical trials. In a randomized, double-blind, placebo-controlled pilot study by Yajima et al. [23] of patients with mild diabetes patients, subjects were supplemented with 80 mg of IAAs or an equivalent placebo twice daily for 12 weeks. This resulted in a significant reduction in blood glucose and hemoglobin A1c (HbA1c) by the eighth week of supplementation. A clinical trial by Obara et al. [29] further clarified the effect of IAAs on subjects with prediabetes. In this randomized, double-blind, dose-finding study, subjects ingested placebo or 16, 32, or 48 mg of IAAs for 12 weeks. After treatment, fasting blood glucose decreased in the 32 and 48 mg groups after 4 weeks, while HbA1c levels decreased in the 16 mg group after 4 weeks and in the 32 and 48 mg groups after 8 weeks. Furthermore, body mass index (BMI) and total fat area significantly decreased in the 48 mg group after 12 weeks. These outcomes revealed the marked benefits of IAAs and their double-agonist action on PPARα and PPARγ.

Given that IAAs exhibit both anti-obesity and anti-diabetic effects via activation of PPARα and PPARγ, IAAs were hypothesized to markedly prevent obesity-induced cognitive decline. Ayabe et al. [30] investigated whether IAA supplementation prevent obesity-induced cognitive decline in the HFD-induced obese mice. A normal diet, HFD, or HFD supplemented with IAAs (0.05%) were fed to C57BL/6 mice for 8 weeks. The HFD increased body weight and visceral fat weight gains, pro-inflammatory cytokine levels in the hippocampus, and impaired episodic memory function assessed via the NORT. IAA supplementation significantly prevented obesity, neuroinflammation, and cognitive decline. Given these findings, IAA supplementation may be effective for the prevention of neuroinflammation and cognitive decline induced by lifestyles.

### 2.3. IAAs Enhance Cognitive Function via Activation of the Vagus Nerve and Dopamine Signaling

To investigate the effects of IAAs on memory function, Ano et al. [31] performed the Y-maze test in a scopolamine-induced amnesia mouse model. This behavioral procedure is designed to assess hippocampus-dependent spatial working memory and has been widely used to screen drugs for dementia efficacy [32,33,34]. Intragastric administration of IAAs (0.2, 2 mg/kg body weight) significantly improved spatial working memory function in a dose-dependent manner. The effect of IAAs was also assessed via the novel object recognition test (NORT), which evaluates hippocampus-dependent episodic memory function [35]. IAAs (0.2, 2 mg/kg) enhanced episodic memory in a dose-dependent manner. According to the allometric scaling to human dose, 0.2–2 mg/kg of IAAs is equivalent to 0.03–0.3 mg/kg (2–20 mg in 70-kg person) in humans. Taniguchi et al. [16] reported that 16–27 mg/L of IAAs are contained in regular types of beer, and thus effective dose of IAAs would be found in 0.13–1.3 L of beer.

In addition to memory function, Ayabe et al. [36] examined the effects of IAAs on prefrontal cortex (PFC)-associated higher-order cognitive functions, such as attention and executive function, using a rodent touch panel operant system. Rodent touch panel operant systems are recently established behavioral devices that offer the advantage of similarity to human cognitive assessment devices and thus good translatability [37]. Using this apparatus, the visual discrimination (VD) task, which assesses the integration of perceptual learning and memory processing [38,39], and the reversal discrimination (RD) task, which assesses cognitive flexibility [40,41], were performed. Repeated administration of IAAs (1 mg/kg) improved cognitive performance on the VD and RD tasks. Treatment with IAAs also enhanced response duration on the VD task, which indicates improved attention.

Although the distinct effects of IAAs on human cognitive function have not been elucidated, one open-label, single-arm, before-and-after design clinical trial was conducted [42]. In this trial, healthy, middle-to-older aged adults were treated with a beverage containing IAAs (3 mg/190 mL) for 4 weeks. A recently-developed assay of brain function based on magnetic resonance imaging (MRI) called the Brain Healthcare Quotient (BHQ) based on gray matter volume (GM-BHQ) [43], was significantly improved after the intervention compared to baseline. This preliminary data suggested that IAA treatment effectively modulated human brain activity.

Hippocampal dopamine (DA) signaling, especially the activation of DA D1-like receptor subtypes (D1 and D5 receptors), plays essential roles in spatial working and episodic memory [44,45,46]. Focusing on hippocampal dopamine neurotransmission, mechanisms underlying the effect of IAAs were investigated in a report by Ano et al. Intragastric treatment with IAAs (0.6 mg/kg) significantly increased hippocampal DA and its metabolite content. Ingestion of IAAs (0.5 mg/kg) also significantly increased extracellular DA levels in the hippocampus, which were assessed using an in vivo microdialysis system. These observations indicated that treatment with IAAs increased both total and extracellular DA levels in the hippocampus.

To further investigate the involvement of DA receptor activation in the improvement of memory function by IAAs, SCH23390, a D1-like receptor antagonist, was used. Intraperitoneal treatment with SCH23390 attenuated the memory improvement effects of IAAs on the Y-maze test and the NORT, suggesting that D1-like receptors modulate the memory-related function of IAAs. Recently, a brain-region specific knockdown of the D1 receptor subtype was established using the injection of adeno-associated viral vectors expressing artificial microRNA (miRNA), which targets the D1 receptor subtype [47]. Treatment with IAAs did not enhance memory function in these D1 knockdown mice, suggesting that the DA D1 receptors in the hippocampus play important roles in the function of IAAs.

As mentioned above, IAAs are potent agonists for T2Rs, which are abundantly expressed in gastrointestinal enteroendocrine cells [48,49]. Bitter stimuli induce intracellular Ca^2+^ increases in enteroendocrine cells, which subsequently release cholecystokinin (CCK), a GI hormone that transmits signals via CCK receptors and stimulates vagus nerve activity [50]. The vagus nerve communicates visceral stimuli to the locus coeruleus (LC), which is located in the brain stem, and then to various brain regions including the hippocampus. Kempadoo et al. [51] reported that DA release from the LC to the dorsal hippocampus improves hippocampus-dependent spatial memory. Takeuchi et al. [52] reported that the dopaminergic neurons projecting from the LC to the hippocampus mediate episodic memory functions. Given this, the cognitive improvement effects of IAAs are thought to be mediated by the activation of T2Rs and consequent vagus nerve stimulation. Indeed, vagotomy operations attenuated the effect of IAAs on increasing DA content in the hippocampus and improved spatial and episodic memory functions. Taken together, these results indicate that IAAs may activate DA signaling via stimulation of the vagus nerve, which enhances hippocampus-dependent memory function.

## 3. Matured Hop Bitter Acids (MHBAs)

### 3.1. Characterization of MHBAs

α- and β-acids are rapidly oxidized during hop storage. Some oxidized compounds of α-acids, such as humulinones, or those of β-acids, such as hulupones, have been identified, but their occurrence in oxidized hops remained unclear until a study by Taniguchi et al. [53]. These authors developed an analytical and preparative method using high performance liquid chromatography (HPLC) for α-acids, β-acids, and their oxidation products, such as humulinones and hulupones, and described structural changes during the hop oxidation process. Taniguchi et al. [16] further developed a preparative method to fractionate total bitter acid oxides, and designated the oxidized fraction as matured hop bitter acids (MHBAs). They analyzed MHBA components and found that MHBAs are primarily composed of the β-tricarbonyl moiety chemical structure common in IAAs and α- and β-acids (Figure 2). Taniguchi et al. further developed a quantitative method for the assessment of MHBAs and evaluated the total amount of MHBAs in several commercial beers. They found 19–38 mg MHBAs/L in lager-type beers available in Japan. IPA and Lambic, which is manufactured with a long aging period, contain relatively high amounts of MHBAs (152–210 and 100–151 mg/L, respectively). These results further suggest that MHBAs contribute to the characteristic flavors of certain types of beers.

### 3.2. MHBAs Improve Lipid Metabolism and Obesity-Induced Cognitive Decline

Given that MHBAs possess common structure IAAs, it was hypothesized that MHBAs might exhibit similar physiological functions to IAAs and in particular a similar anti-obesity efficacy. To test this, Morimoto-Kobayashi et al. [54] examined the anti-obesity effects of MHBAs in rodents. C57BL/6 mice were fed a control diet, a HFD, and an HFD supplemented with 0.05% (*w*/*w*) MHBAs for 12 weeks. Supplementation with MHBAs significantly reduced body weight gains, epididymal fat weight, and plasma non-esterified free fatty acids as compared to the HFD alone. The authors also found that MHBAs increased the expression of uncoupling protein-1 in the brown adipose tissue (BAT). Further, Morimoto-Kobayashi et al. showed that an acute, single administration of MHBAs enhanced thermogenesis in the BAT via sympathetic nerve activity stimulation, which was blocked by a vagotomy. Yamazaki et al. [55] elucidated the mechanism underlying MHBA effects on the sympathetic nervous system. Treatment of enteroendocrine cells with MHBAs increased Ca^2+^ intracellular influx and CCK secretion, which was eliminated by Ca^2+^ depletion or treatment with an L-type voltage-sensitive Ca^2+^ channel blocker. In animal experiments, sympathetic nerve activity innervating BAT (BAT-SNA) and BAT temperature increases with MHBA treatment was attenuated by a CCK receptor 1 antagonist while intraperitoneal injection of CCK fragment peptides enhanced BAT-SNA. Serial observations suggested that MHBAs stimulated enteroendocrine cells and induced Ca^2+^ influx and CCK secretion, which increases BAT-SNA and BAT thermogenesis via the CCK receptor 1 and the vagus nerve.

The anti-obesity effects of MHBAs in humans were investigated in a randomized, double-blind, placebo-controlled study [56,57]. Two-hundred subjects (aged 20–65 years) with BMIs of 25–30 kg/m^2^ were randomly assigned to two groups and treated with MHBAs (35 mg/kg) or placebo for 12 weeks. Treatment with MHBAs significantly reduced visceral fat areas, as evaluated by CT scans at weeks 8 and 12, and the total fat area at week 12, indicating that MHBAs reduced fat accumulation in humans. This study suggested that continuous ingestion of MHBAs reduces body fat of healthy overweight subjects.

Taking account of anti-obese effects of MHBAs, Ayabe et al. [58] investigated the effects of MHBAs on obesity-induced cognitive decline. C57BL/6 mice were fed with normal diet, HFD, or HFD supplemented with MHBAs (0.05%) for 8 weeks. Dietary MHBA supplementation administered to HFD-fed mice reduced body weight and epididymal fat weight, and improved episodic memory deficits. The authors also found the negative correlation between the score of episodic memory and both the body weight and epididymal fat weight, and thus concluded that prevention of obesity by MHBA supplementation would be effective in prevention of obesity-induced cognitive decline.

### 3.3. MHBAs Enhance Cognitive Function via Activation of the Vagus Nerve and Norepinephrine Signaling

The effects of MHBAs on memory function were first revealed by Ayabe et al. [59]. Intragastric treatment with MHBAs (10 mg/kg) enhanced spatial working memory in a scopolamine-induced amnesia mouse model in the Y-maze test and episodic memory in normal mice in the NORT. The effect of several representative components in MHBAs were examined, and 4′-hydroxyallohumulinone (HAH; 0.1, 1 mg/kg) and 4′-hydroxy-*cis*-alloisohumulone (HAIH; 1 mg/kg) were found to enhance spatial working memory in the Y-maze test.

Furthermore, MHBAs were previously reported to stimulate the vagus nerve [54]. The vagus nerve mediates visceral stimuli to the LC. The LC communicates with various brain regions via noradrenergic neurons, and activation of LC-noradrenergic pathways improves object recognition [60,61]. The effects of MHBAs on hippocampal NE signaling were further examined, and MHBAs were found to increase NE content and extracellular NE levels in the hippocampus. Furthermore, given the attenuation of the memory enhancement effects of MHBAs with propranolol, a β-adrenergic receptor antagonist, NE signaling modulates MHBA effects. Further, vagotomy operations diminished the effects of MHBAs in the Y-maze test and the NORT. This report concluded that MHBAs enhanced memory function via stimulation of the vagus nerve via enhanced NE signaling.

Previous studies have shown that stimulation of the vagus nerve activates the central cholinergic system [62,63]. The neurotransmitter acetylcholine (ACh) plays essential roles in memory function, especially in episodic or spatial memories [64]. Focusing on the cholinergic system, Fukuda et al. [65] further investigated the mechanisms underlying MHBA function. They used several ACh receptor (AChR) antagonists and found that nicotinic AChR, in particular, is involved in the memory improvements following MHBA supplementation.

Recently, Fukuda et al. [66] performed a randomized, double-blind placebo-controlled clinical trial to investigate the effects of MHBAs on human cognitive function. Sixty healthy adults (aged 45–64 years) with insight into their own cognitive decline were randomly divided into two groups and supplemented with MHBAs (35 mg/kg) or placebo for 12 weeks. Their cognitive performance was evaluated via a neuropsychological battery at weeks 0 (baseline), 6, and 12. Changes in participant verbal fluency test scores at week 6 were significantly more improved from baseline in the MHBA-supplemented group than in the placebo group. Stroop test score changes at week 12 were also significantly improved from baseline in the MHBA group. The verbal fluency test evaluates memory retrieval function while the Stroop test evaluates executive function. Both tests are closely related to PFC functions [67,68]. These results suggest that supplementation with MHBAs may be effective for the maintenance of cognitive function. Fukuda et al.’s clinical trial was the first report to demonstrate the effectiveness of hop-derived bitter acids on human cognitive function [66]. Despite this, however, the mechanism(s) underlying MHBA effects on PFC function have not been elucidated, and further studies are required.

In the clinical trial of Fukuda et al., the effect of MHBAs on mental fatigue and mood states were investigated as secondary outcomes. Treatment with MHBAs (35 mg/day) for 12 weeks significantly relieved subjective fatigue, as assessed via the visual analogue scale, and tension-anxiety state features, as assessed by the Profile of Mood States 2. These results suggested that MHBAs might improve both memory and psychiatric functions in humans.

Fukuda et al. [69] further investigated the effects of MHBAs on psychiatric functions in animal models. Intragastric treatment with MHBAs (10 and 50 mg/kg) increased NE levels and suppressed depression-like behaviors, as assessed by the tail suspension test, in a mouse model of depression. HAH and HAIH, representative components of MHBAs, which modulate memory improvements also had anti-depression effects. The authors further reported that NE increase and anti-depression effect of MHBAs were attenuated by vagotomy operation. These observations suggest that treatment with MHBAs suppresses depression-like behaviors via activation of the vagus nerve and NE signaling, findings that are consistent with the mechanisms underlying memory enhancement.

IAAs enhance cognitive function via the activation of the vagus nerve and DA neurotransmissions. Similarly, MHBAs improve memory functions via the activation of vagus nerve and NE or ACh neurotransmission (Figure 3). IAAs and MHBAs both activate the vagus nerve, but the consequent responses in the brain differ, presumably depending on the combination of T2Rs activated by these bitter acids. A previous study reported that oxidized bitter components elicit milder bitterness than those elicited by IAAs [70], suggesting that IAAs and MHBAs differently impact T2Rs. Further studies to identify the receptor subtypes for MHBAs and their individual components may provide new approaches for modifying brain neurotransmission by bitter components.

Several studies have reported that vagus nerve stimulation has beneficial effects on cognitive functions [71,72,73]. However, established procedures to stimulate the vagus nerve require surgery to implant electrophysiological devices, implicating safety and complication risks. IAAs have been consumed for more than a thousand years, which may support the safety of IAA intake. Suzuki et al. [74] conducted the in vitro and in vivo safety study of MHBAs, and reported no safety concern even in the overdoses. Given this, daily hop-derived bitter acid supplementation may be a safer and easier strategy to stimulate the vagus nerve and subsequently brain neurotransmission, and thus to improve cognitive functions.

## 4. Potential for Hop-Derived Bitter Acids in Alzheimer’s Disease Treatment

AD is the most common type of dementia and is characterized by extracellular amyloid plaques, which are deposits of amyloid β (Aβ) [75], and intracellular neurofibrillary tangles of hyperphosphorylated tau proteins [76]. Further, inflammation in the brain induced by Aβ deposition plays a major role in the neurodegenerative pathology underlying AD [77]. Neuroinflammatory changes in AD are associated with microglia activation. Microglia are the innate immune cells of the central nervous system and exhibit two phenotypes: M1 (pro-inflammatory) and M2 (anti-inflammatory). Recently, NOD-like receptor (NLR) family pyrin domain containing 3 (NLRP3) inflammasomes inside microglia have been found to play roles in the neuroinflammatory processes of AD [78]. The activation of NLRP3 induces the release of pro-inflammatory cytokines such as interleukin (IL)-1β and IL-18. Another recent study found that DA inhibits the pro-inflammatory NLRP3 inflammasome response via the activation of the DA D1 receptor [79]. Inflammation in the brain is critical to neuronal degeneration and thus suppression of neuroinflammation may be an effective approach to AD prevention.

Activation of PPARγ induces the anti-inflammatory state transformation of microglia. Given this, PPARγ may be a fruitful therapeutic target for AD [80]. IAAs, which are potent agonists of PPARγ, may have preventive effects on AD. These effects have been investigated using various mouse models of AD or cognitive decline. For instance, using 5xFAD mice, a mouse model of familial AD which exhibits massive Aβ deposition and cognitive decline, Ano et al. [81] revealed the preventive effects of IAAs on AD-like pathologies. The dietary supplementation of IAAs (0.05% *w*/*w*) for 3 months suppressed Aβ deposition and levels of pro-inflammatory cytokines, such as IL-1β, and chemokines, such as macrophage inflammatory protein (MIP)-1α. To evaluate cognitive function, the authors used the NORT and found that the supplementation with IAAs prevented cognitive impairments in the AD mouse model. IAAs were further confirmed to induce an anti-inflammatory state in microglia, suppressing inflammatory responses and enhancing Aβ phagocytosis via the activation of PPARγ.

The dementia prevention effects of IAAs were also investigated in the rTg4510 mouse strain, a model of tauopathy [82]. Consumption of IAAs for 3 months significantly reduced levels of pro-inflammatory cytokines and chemokines including IL-1β and MIP-1α, respectively, and reduced the population of microglia producing these cytokines and chemokines. Notably, the consumption of IAAs reduced levels of phosphorylated tau in the brain. A recent report revealed that the activation of microglia and the NLRP3 inflammasome is involved in tauopathy [83], and thus IAAs may influence tau-related pathologies via anti-inflammatory modulation.

Hippocampal hyperactivity is another characteristic pathology of AD, and hyperactivity suppression has been reported to improve cognitive impairment [84]. Supplementation with IAAs suppressed neuroinflammation and AD-related hyperactivity in the hippocampi of J20 mice, a model of familial AD, assessed via rodent MRI [85]. This study also examined the short-term effects of IAAs and found that intragastric administration of IAAs for 7 days attenuated neuroinflammation and cognitive decline in 5xFAD mice, suggesting the therapeutic potential of IAAs after AD onset. Potent agonistic activity of IAAs on PPARγ may further contribute to the preventive effects of AD via suppression of neuroinflammation. Moreover, DA or specific D1 receptor agonists inhibit NLPR3-mediated inflammation via D1 receptor activation [79,86]. Enhancement of DA content and activation of DA D1 receptors with IAA treatment may thus contribute to the AD-prevention effects of IAA.

In addition to studies using genetically modified AD mouse models, more spontaneous models of neuroinflammation and cognitive decline, such as aged mice, have also been used to examine the effect of IAAs. To evaluate the effects of IAAs on age-related neuroinflammation and cognitive decline, aged mice (68 weeks of age) were supplemented with IAAs (0.05%) for 3 months [87]. As compared to younger mice (7 weeks of age), aged mice exhibited significantly decreased DA levels and spatial working and episodic memory functions, and increased pro-inflammatory microglial phenotypes, pro-inflammatory cytokine and chemokine levels, and Aβ levels. Supplementation with IAAs attenuated these age-related changes in memory function, DA levels, inflammatory states, and Aβ levels. Short-term administration of IAAs for 7 days also improved age-related cognitive decline in the aged mice.

The LC is thought to play important roles in the progression of AD. Degeneration of the LC and noradrenergic neurons projecting from the LC has been observed in most AD patients [88]. Heneka et al. [89] reported that the degeneration of the LC induced Aβ-related pathologies in the mouse model of AD. They also reported that the loss of LC neurons induced microglial inflammation and dysfunction, and that supplementation with an NE precursor rescued AD-like pathologies in AD model mice [90]. Further, Vonck et al. [91] elucidated the therapeutic potential of vagus nerve stimulation for AD, suggesting the critical involvement of NE and its anti-inflammatory effects. Given these reports, MHBAs, which activate the vagus nerve and NE signaling in the brain, may prevent or treat AD and dementia. Studies investigating the effects of MHBAs on AD prevention have not yet been conducted, though they are anticipated.

## 5. Conclusions

Recent findings suggest robust memory-improving effects of hop-derived bitter acids. IAAs, the main bitter components of beer, improve hippocampus-dependent memory and PFC-associated cognitive functions. These functions are mediated by increased DA levels and the activation of dopamine D1 receptors. MHBAs, bitter acid oxides with a β-carbonyl structure, also enhance rodent memory functions via activation of NE signaling, and improve human cognitive and psychiatric functions. Notably, the cognitive improvement effects of IAAs and MHBAs are mediated by vagus nerve stimulation. Moreover, IAAs exhibit preventive effects on dementia and cognitive decline, which may be mediated by the anti-inflammatory processes driven by PPARγ activation. Previous reports have estimated that reasonable amounts of beer (containing 0.13–1.3 L of IAAs and 0.17–1.8 L of MHBAs) are sufficient to ingest effective doses of these compounds in humans [31,66]. These compounds may contribute to the dementia prevention effects of alcoholic beverages including beer, and it is expected that certain types of beer brewed with a large amount of hops might be more beneficial. Daily intake of hop-derived bitter acids may, thus, be beneficial for the maintenance of cognitive function.

## Figures and Tables

**Figure 1 biomolecules-10-00131-f001:**
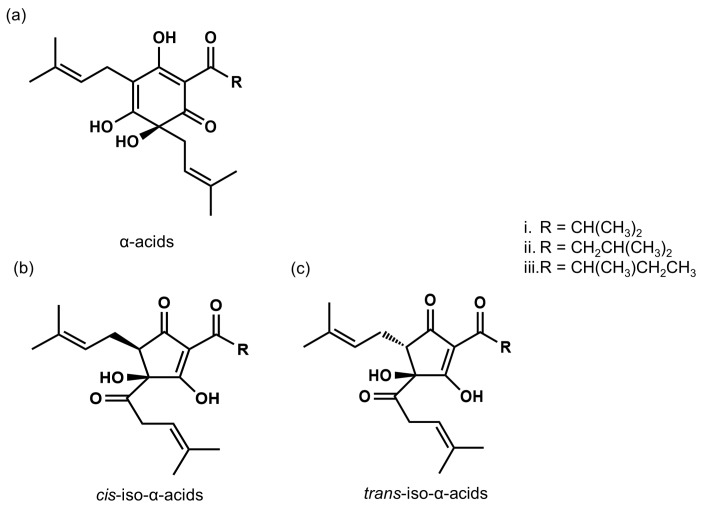
The chemical structures of α-acids and iso-α-acids. (**a**) α-acids; cohumulone (i), *n*-humulone (ii), and adhumulone (iii). (**b**) *cis*-iso-α-acids; *cis*-isocohumulone (i), *cis*-isohumulone (ii), and *cis*-isoadhumulone (iii). (**c**) trans-iso-α-acids; trans-isocohumulone (i), trans-isohumulone (ii), and trans-isoadhumulone (iii).

**Figure 2 biomolecules-10-00131-f002:**
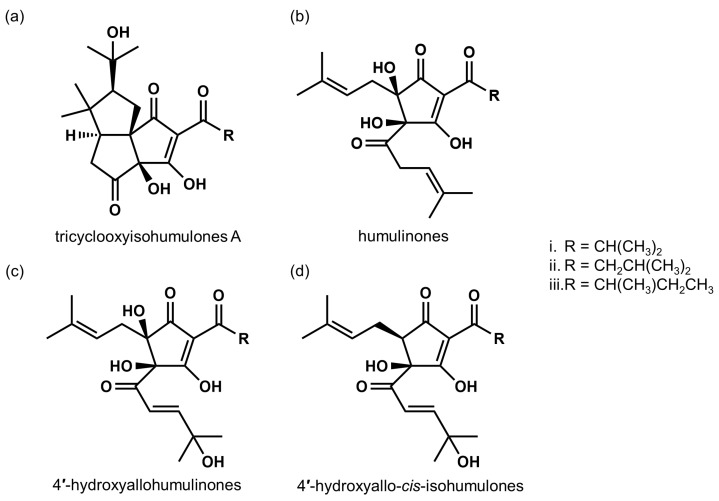
Chemical structures of representative MHBA components. (**a**) tricyclooxyisohumulones A; tricyclooxyisocohumulone A (i), tricyclooxyisohumulone A (ii), and tricyclooxyisoadhumulone A (iii). (**b**) humulinones; cohumulinone (i), humulinone (ii), and adhumulinone (iii). (**c**) 4′-hydroxyallohumulinones; 4′-hydroxyallocohumulinone (i), 4′-hydroxyallohumulinone (ii), and 4′-hydroxyalloadhumulinone (iii). (**d**) 4′-hydroxyallo-*cis*-isohumulones; 4′-hydroxyallo-*cis*-isocohumulone (i), 4′-hydroxyallo-*cis*-isohumulone (ii), 4′-hydroxyallo-*cis*-isoadhumulone (iii).

**Figure 3 biomolecules-10-00131-f003:**
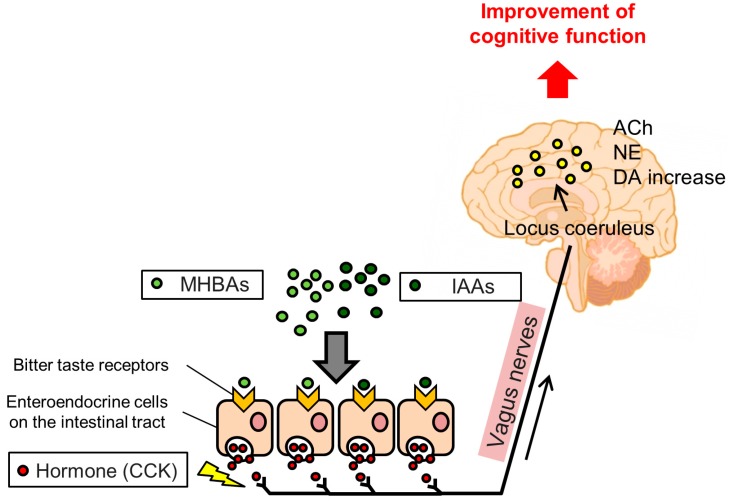
Expected mechanism underlying the memory improvement effects of Iso-α-acids (IAAs) and Matured hop bitter acids (MHBAs). IAAs and MHBAs may bind to bitter taste receptors in the enteroendocrine cells in the intestinal tract, subsequently inducing cholecystokinin (CCK) release, which stimulates the vagus nerve. Activation of the vagus nerve by IAAs and MHBAs enhances dopamine (DA), norepinephrine (NE,) and acetylcholine (ACh) increases, which may improve cognitive function.

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
