# Peer review of "Improving Effects of Hop-Derived Bitter Acids in Beer on Cognitive Functions: A New Strategy for Vagus Nerve Stimulation"

_biomolecules, 2020, doi:10.3390/biom10010131_

Round 1
Reviewer 1 Report
The manuscript entitled "Improving effects of hop-derived bitter acids in beer on cognitive functions: A new strategy for vagus nerve stimulation" outlines the impacts of bitter acids from hops on cognitive function. The manuscript is well written with only a few corrections required for publication.
In general, the authors should address the link between obesity and cognitive function in the introduction. It will then set the stage for their discussion on bitter acids impacts on metabolic disorder and obesity. Currently the link between obesity and cognitive function is not outlined until line 341.
Line 80. "... and adhumulone (iii) ...
Line 244. Provide reference for Fukuda at al.
Line 341 "... obesity induce neuroinflammation, ..."
References.
The authors should refer to the Guidelines for Authors to insure that the formatting meets all the requirements for references.
Line 402. Reference 11 is missing.
Author Response
The manuscript entitled "Improving effects of hop-derived bitter acids in beer on cognitive functions: A new strategy for vagus nerve stimulation" outlines the impacts of bitter acids from hops on cognitive function. The manuscript is well written with only a few corrections required for publication.
In general, the authors should address the link between obesity and cognitive function in the introduction. It will then set the stage for their discussion on bitter acids impacts on metabolic disorder and obesity. Currently the link between obesity and cognitive function is not outlined until line 341.
Response:
We appreciate your careful reading and the valuable comments. We have mentioned the relationship between obesity and dementia in the introduction.
Line 80. "... and adhumulone (iii) ...
Line 244. Provide reference for Fukuda at al.
Line 341 "... obesity induce neuroinflammation, ..."
Response:
Thank you for pointing these out. We have corrected these points.
References.
The authors should refer to the Guidelines for Authors to insure that the formatting meets all the requirements for references.
Line 402. Reference 11 is missing
Response:
We have checked the guidelines for references and formatted.

Reviewer 2 Report
In this review manuscript, authors reported that hop-derived bitter acids help to improve cognitive function. They targeted two different ingredients IAAs and MHBAs of bitter acids, and discussed recent findings of memory-improving effects. In the scheme figure authors explained that these compounds enhance cognitive function mainly via vagus nerve stimulation, and then upregulate some brain neurotransmitters secretion including DA, NE, and ACh. Therefore, supplementation with IAAs and MHBAs may attenuate neuroinflammation and cognitive impairments. This manuscript is a detailed review of the relationship between hop-derived bitter acids and cognitive functions. I think it valuable to be accepted for publication in this journal, and some minor comments are attached for the author's reference
2.2: I think the entire paragraph is mainly describing the relationship between IAAs and T2D. However, this part seems to not mention any narratives related to cognitive function changes. Since T2D is known to be one of the main risks of cognitive impairments such as Alzheimer's disease, I suggest authors may discuss and include the above points in this paragraph to echo its title. 2.3: There is too little narrative about the vagus nerve, and how the IAAs affect the possible mechanism of the vagus nerve is also not explained. 3.3 It is very interesting to state that MHBAs (and IAAs) stimulate the vagus nerve to increase the secretion of DA, NE, and ACh in the brain, thereby protecting the cognitive function. Compared to vagus nerve stimulation by implanting electrophysiological devices, oral MHBAs and IAAs are indeed safer. However, too much dose of MHBAs and IAAs can still have some side effects and even harmful. Can authors provide some information on this and include it in the text? “4. Potential for hop-derived bitter acids in disease treatment”: This section is almost all about Alzheimer's disease. Maybe it would be more understandable to change the "disease" of the subtitle directly to “Alzheimer's disease”.
Author Response
In this review manuscript, authors reported that hop-derived bitter acids help to improve cognitive function. They targeted two different ingredients IAAs and MHBAs of bitter acids, and discussed recent findings of memory-improving effects. In the scheme figure authors explained that these compounds enhance cognitive function mainly via vagus nerve stimulation, and then upregulate some brain neurotransmitters secretion including DA, NE, and ACh. Therefore, supplementation with IAAs and MHBAs may attenuate neuroinflammation and cognitive impairments. This manuscript is a detailed review of the relationship between hop-derived bitter acids and cognitive functions. I think it valuable to be accepted for publication in this journal, and some minor comments are attached for the author's reference
We appreciate your careful reading and the valuable comments.
2.2: I think the entire paragraph is mainly describing the relationship between IAAs and T2D. However, this part seems to not mention any narratives related to cognitive function changes. Since T2D is known to be one of the main risks of cognitive impairments such as Alzheimer's disease, I suggest authors may discuss and include the above points in this paragraph to echo its title.
Thank you for your comment. Following your suggestion, we have described the relationship between dementia and type 2 diabetes or obesity. We also showed the preventive effect of IAAs on obesity-induced cognitive decline in this section.
2.3: There is too little narrative about the vagus nerve, and how the IAAs affect the possible mechanism of the vagus nerve is also not explained.
We have added the information about the vagus nerve, and mentioned that IAAs may activate gut bitter taste receptor and then stimulate vagus nerve.
3.3 It is very interesting to state that MHBAs (and IAAs) stimulate the vagus nerve to increase the secretion of DA, NE, and ACh in the brain, thereby protecting the cognitive function. Compared to vagus nerve stimulation by implanting electrophysiological devices, oral MHBAs and IAAs are indeed safer. However, too much dose of MHBAs and IAAs can still have some side effects and even harmful. Can authors provide some information on this and include it in the text?
Our group previously performed the safety study on MHBAs, and reported that the acute toxicity level of MHBAs is higher than 2,000 mg/kg BW, and the no-observed-adverse-effect-level was over 3,484 mg/kg BW. Thus there would not be any safety concern on MHBAs even in overdoses.
On the other hand, the safety study of IAAs is not reported so far. However, IAAs have been consumed over 1,000 years, which may support the safety of IAAs.
These information are now added to the article.
“4. Potential for hop-derived bitter acids in disease treatment”: This section is almost all about Alzheimer's disease. Maybe it would be more understandable to change the "disease" of the subtitle directly to “Alzheimer's disease”.
Thank you for pointing this out. We changed the subtitle.

Reviewer 3 Report
The article deals with hop-derived bitter acids on cognitive functions. The study is relevant to the field of development of functional food and beverages. There are few basic flaws however, which must be fixed before the manuscript could be considered for publication. First, Authors do not show clearly if the results effects of hop-derived bitter acid on cognitive functions were obtained by using beer containing the biomolecules active or by using only of hop-derived bitter acid.
Abstract: Line 24, If possible, please authors should insert the quantity of the daily supplementation (for example, from 1 to 3 g) or beer dose/day.
Keywords: Please, could authors insert β-carbonyl as a keyword…
Introduction: Line 62-63, Please, if possible, authors should report if the use of xanthohumol concentrated for beer production can lead to ingesting an effective dose with only moderate beer consumption........
Line 64-68, I think that authors should include the importance to obtain a beer with effective dose de biocompounds with only moderate beer consumption.......
"2.2. IAAs improve type II diabetes and lipid metabolism via PPAR activation" .....could Authors focus on the effects of hop-derived bitter acids, especially on cognitive function. Information reported seems not to contribute to the goal and title of the present paper...
Line 119, Please could you explain if it can be found 0.2,2 mg/kg in 1-2 drinks/day……verify Taniguchi et al. [15]……
Line 213-233, The authors should have focused on the effects of MHBAs on cognitive function.......
Conclusion: Line 372-374, Authors should explain how can found hop-derived bitter acids, IAAs and MHBAs…….. in wine? or delete wine... Please, If possible, should report if the beer type consumed increases the beneficial for the maintenance of cognitive function.
Figures: If any of the figures you have included are unoriginal, copyright permission for their use will need to be obtained from the original publishers. Most major publishing companies have instructions on how to obtain permission on their websites, usually with an online request system.
Author Response
The article deals with hop-derived bitter acids on cognitive functions. The study is relevant to the field of development of functional food and beverages. There are few basic flaws however, which must be fixed before the manuscript could be considered for publication. First, Authors do not show clearly if the results effects of hop-derived bitter acid on cognitive functions were obtained by using beer containing the biomolecules active or by using only of hop-derived bitter acid.
Response:
Thank you for your careful reading and the valuable comments. All studies of hop-derived bitter acids are performed using hop extract containing the bitter acids, not beer. These points are now described in the introduction.
Abstract: Line 24, If possible, please authors should insert the quantity of the daily supplementation (for example, from 1 to 3 g) or beer dose/day.
Response:
We added an example of the quantity of bitter acid supplementation.
Keywords: Please, could authors insert β-carbonyl as a keyword…
Response:
We added β-carbonyl as a keyword.
Introduction: Line 62-63, Please, if possible, authors should report if the use of xanthohumol concentrated for beer production can lead to ingesting an effective dose with only moderate beer consumption........
Response:
We have mentioned that it is now possible to concentrate xanthohumol to produce functional food.
Line 64-68, I think that authors should include the importance to obtain a beer with effective dose de biocompounds with only moderate beer consumption.......
Response:
We have mentioned the importance to achieve effective dose with only moderate beer consumption, because of the harmful effects of excessive alcohol consumption.
"2.2. IAAs improve type II diabetes and lipid metabolism via PPAR activation" .....could Authors focus on the effects of hop-derived bitter acids, especially on cognitive function. Information reported seems not to contribute to the goal and title of the present paper...
Response:
Thank you for your comment. Other reviewers also mentioned this, and suggested that the relationship between type II diabetes and cognitive decline should be discussed in this section. Following this suggestion, we have described the effect of IAAs on obesity-induced cognitive decline. I hope this change would be satisfactory for your comment.
Line 119, Please could you explain if it can be found 0.2,2 mg/kg in 1-2 drinks/day……verify Taniguchi et al. [15]……
Response:
We mentioned that the effective dose of IAAs (0.2,2 mg/kg in mice) would be found in 0.13-1.3 L of beer.
Line 213-233, The authors should have focused on the effects of MHBAs on cognitive function.......
Response:
Similar to the above comment, we have discussed the effects of MHBAs on obesity-induced cognitive decline in this section.
Conclusion: Line 372-374, Authors should explain how can found hop-derived bitter acids, IAAs and MHBAs…….. in wine? or delete wine... Please, If possible, should report if the beer type consumed increases the beneficial for the maintenance of cognitive function.
Response:
We deleted “wine”.We also mentioned that certain types of beer brewed with large amount of beer may be more effective.
Figures: If any of the figures you have included are unoriginal, copyright permission for their use will need to be obtained from the original publishers. Most major publishing companies have instructions on how to obtain permission on their websites, usually with an online request system.
Response:
The figures attached are original.
